# Resolving the topology of encircling multiple exceptional points

Chitres Guria[1], Qi Zhong[2,3], Sahin Kaya Ozdemir [3], Yogesh S. S. Patil [1], Ramy El-Ganainy [2,4] ✉ & Jack Gwynne Emmet Harris [1,5,6] ✉

Non-Hermiticity has emerged as a new paradigm for controlling coupled-mode systems in ways that cannot be achieved with conventional techniques. One aspect of this control that has received considerable attention recently is the encircling of exceptional points (EPs). To date, most work has focused on systems consisting of two modes that are tuned by two control parameters and have isolated EPs. While these systems exhibit exotic features related to EP encircling, it has been shown that richer behavior occurs in systems with more than two modes. Such systems can be tuned by more than two control parameters, and contain EPs that form a knot-like structure. Control loops that encircle this structure cause the system's eigenvalues to trace out non-commutative braids. Here we consider a hybrid scenario: a three-mode system with just two control parameters. We describe the relationship between control loops and their topology in the full and two-dimensional parameter space. We demonstrate this relationship experimentally using a three-mode mechanical system in which the control parameters are provided by optomechanical interaction with a high-finesse optical cavity.

Modeling physical systems as collections of coupled oscillators is an important strategy in many disciplines, including quantum field theory, condensed matter, optics, acoustics, soft matter, and biology. Classical linear coupled-oscillator models (COMs) are described by the system of equations $\dot{\mathbf{x}} = -iH\mathbf{x}$ where the complex $N$-vector $\mathbf{x}$ represents the phase space coordinates of $N$ oscillators, and the $N \times N$ matrix $H$ encodes the oscillators' properties. One important feature of any COM is its spectrum of resonance frequencies $\boldsymbol{\lambda}$, which is the (unordered) set of the eigenvalues of $H$. In many applications, it is desirable to tune $\boldsymbol{\lambda}$ by varying parameters that appear in $H$. This tuning may be static, in the sense of setting the parameters in order to fix specific resonances (e.g., when using the COM as a transducer for external signals with known frequencies). Alternatively, the tuning may be dynamical in the sense of being carried out in real time to manipulate excitations within the COM (as in adiabatic control schemes). In both cases, it is important to understand how $\boldsymbol{\lambda}$ depends upon parameters in

$H$. While the details of this dependence will vary from one system to another, it is known to possess a number of generic features.

One such feature that has attracted considerable attention recently is the topological structure that emerges when $H$ is non-Hermitian[1-6]. This structure is manifested in the evolution of $\boldsymbol{\lambda}$ when the parameters of $H$ are varied around a loop[7-13]. The simplest example occurs in a system of $N = 2$ oscillators with two control parameters ($b_1$ and $b_2$) chosen so that the 2D space they span contains a single point at which the eigenvalues are degenerate (for generic $b_1$ and $b_2$ this degeneracy will be an exceptional point (EP)[14,15]). If $\mathbf{b} = (b_1, b_2)$ is varied smoothly around a loop (which does not intersect the EP), the evolution of $\boldsymbol{\lambda}$ will result in a permutation of the two eigenvalues if and only if the loop's winding number around the EP is odd.

For $N > 2$, a generic 2D control space (which we refer to as $\mathcal{B}$) will contain isolated twofold EPs that correspond to degeneracies between various mode pairs[14,15]. In such systems, the relationship between a

[1]Department of Physics, Yale University, New Haven, CT 06520, USA. [2]Department of Physics, Michigan Technological University, Houghton, MI 49931, USA. [3]Department of Engineering Science and Mechanics, and Materials Research Institute, The Pennsylvania State University, University Park, PA 16802, USA. [4]Henes Center for Quantum Phenomena, Michigan Technological University, Houghton, MI 49931, USA. [5]Department of Applied Physics, Yale University, New Haven, CT 06520, USA. [6]Yale Quantum Institute, Yale University, New Haven, CT 06520, USA. ✉e-mail: ganainy@mtu.edu; jack.harris@yale.edu

control loop, the EPs, and the resulting eigenvalue permutation is less intuitive than for $N = 2$. In particular, there is no simple correspondence between a loop's topology (more precisely, its homotopy class in $\bar{\mathcal{B}}$, defined as $\mathcal{B}$ with the EPs removed) and the resulting eigenvalue permutation. For example, loops that are homotopy equivalent in $\bar{\mathcal{B}}$ all give the same permutation, but homotopy inequivalent loops do not necessarily give distinct permutations.

As shown in ref. [16], the permutation associated with a given loop can be calculated by introducing into $\mathcal{B}$ branch cuts (BCs) for $\boldsymbol{\lambda}$ and tracking the manner in which the loop crosses these BCs, and the resulting eigenvalue permutation is less intuitive than for $N = 2$[16,17]. This approach has the advantage of being straightforward to visualize, as it only involves quantities that are defined in the 2D space $\mathcal{B}$. It is also relevant to the many systems that in practice offer just two control parameters. However, relying on the introduction of BCs can obscure the topological relationship between control loops and EPs.

A different approach is to view the control loop in the space that is spanned by all the coefficients of $p_H$, the characteristic polynomial of $H$[18]. These coefficients are simple functions of the elements of $H$, and they provide a smooth parametrization of the eigenvalue spectra[19]. If we take $H$ to be traceless (see "Methods"), then $p_H$ has $N-1$ coefficients. As these are complex, the space they span (which we denote as $\mathcal{L}_N$) is isomorphic to the Euclidean space $\mathbb{R}^{2(N-1)}$.

Within $\mathcal{L}_N$, the degeneracies comprise a subspace (which we denote as $\mathcal{V}_N$) having two dimensions fewer than $\mathcal{L}_N$. Intuitively, this follows because degeneracy corresponds to $\lambda_i = \lambda_j$ (i.e., equality of two eigenvalues), which can be regarded as two real constraints. For a loop $\mathcal{C}$ that does not intersect $\mathcal{V}_N$ (i.e., $\mathcal{C}$ lies entirely in the space of non-degenerate spectra $\mathcal{G}_N \equiv \mathcal{L}_N - \mathcal{V}_N$), it is straightforward to show that the resulting evolution of $\boldsymbol{\lambda}$ is a braid of $N$ strands, where each strand of the braid represents the evolution of one of the eigenvalues of $H$. Furthermore, there is a one-to-one correspondence between the topology of the control loop in $\mathcal{G}_N$ and the topology of the resulting eigenvalue braid (formally, the correspondence is between homotopy equivalence classes of based loops in $\mathcal{G}_N$ and isotopy equivalence classes of braids)[18,20,21].

The high dimension of these spaces and the nontrivial geometry of $\mathcal{V}_N$ (for $N > 2$) make it challenging to visualize this approach. However, it has the advantage of giving the topological character of the evolution of $\boldsymbol{\lambda}$ directly in terms of the manner in which $\mathcal{C}$ encircles the EPs.

The rich behavior exhibited by COMs with $N > 2$ has been shown to offer considerable promise for a range of applications, including enhanced sensing, topological control, and line-narrowing in lasers[22–36]. As a result, it is important to develop an intuitive description of how $\boldsymbol{\lambda}$ can be tuned in such systems. Here, we describe experiments that use a system of $N = 3$ mechanical oscillators to elucidate the connections between viewing $\boldsymbol{\lambda}$ using a 2D control space ($\mathcal{B}$) and using its full control space ($\mathcal{L}_3$). Specifically, we measure $\boldsymbol{\lambda}$ in various $\mathcal{B}$ throughout $\mathcal{L}_3$, and use this data to track the evolution of $\boldsymbol{\lambda}$ around various loops in each $\bar{\mathcal{B}}$. This data also provides the structure of $\mathcal{G}_3$ and $\mathcal{V}_3$, and so allows us to illustrate a number of qualitative features in the topological behavior of $\boldsymbol{\lambda}$. These measurements demonstrate that loops which encircle different EPs in a given $\bar{\mathcal{B}}$ produce the same permutation if they are homotopy equivalent in $\mathcal{G}_3$. They also demonstrate that loops can encircle the same EPs in a given $\bar{\mathcal{B}}$ and produce different permutations if they are homotopy inequivalent in $\mathcal{G}_3$. Lastly, they demonstrate the role of the loops' basepoints in determining their homotopy equivalence.

## Results
### Three-mode systems
To illustrate the view of control loops and EPs provided by the full control space $\mathcal{L}_N$, we give an explicit description of the case $N = 3$. The characteristic polynomial of any $3 \times 3$ traceless matrix can be written as

$p_H = \lambda^3 - y\lambda - x$ where the complex numbers $x$ and $y$ are the control parameters. The space they span is $\mathcal{L}_3$, which is isomorphic to $\mathbb{R}^4$. This means that $\mathcal{V}_3$, the subspace consisting of the degeneracies, will be two-dimensional (as degeneracies comprise a subspace with two fewer dimensions than the full control space).

To find the structure of $\mathcal{V}_3$, we use the fact that a polynomial's roots are degenerate if and only if its discriminant vanishes. For $p_H$, this corresponds to the condition

$$4y^3 = 27x^2 \tag{1}$$

so that the solutions of Eq. (1) are the coordinates of the degeneracies in $\mathcal{L}_3$. The trivial solution $x = y = 0$ (corresponding to the origin of $\mathcal{L}_3$) corresponds to a three-fold degeneracy (which we denote as EP$_3$), with $\boldsymbol{\lambda} = \{0, 0, 0\}$. This is the only EP$_3$ in $\mathcal{L}_3$ (since Tr[$H$] = 0), so the remainder of $\mathcal{V}_3$ must consist of twofold degeneracies (which we denote as EP$_2$ or simply EP). These can be found by first considering a hypersphere $\mathcal{S}_r$ with radius $r$ and centered at the origin (i.e., it is defined by $|x|^2 + |y|^2 = r^2$). It is straightforward to show that this constraint together with Eq. (1) fixes $|x|$ and $|y|$ while also requiring that $2 \arg(x) = 3 \arg(y)$ (here arg denotes the complex argument). This defines a (2, 3) torus knot $\mathcal{K}$ (also known as a trefoil knot) within $\mathcal{S}_r$[37]. Since this reasoning holds for any $r > 0$, the full space of degeneracies $\mathcal{V}_3$ can be viewed as the result of "extruding" $\mathcal{K}$ in the radial direction (i.e., so that it collapses to a point at the origin). The resulting two-dimensional surface is known as the topological cone of the trefoil knot, denoted as $C\mathcal{K}$.

In addition to giving the structure of $\mathcal{V}_3$, this reasoning also provides the structure of $\mathcal{G}_3$ (the non-degenerate space in which control loops are assumed to lie) as this is simply the complement of $\mathcal{V}_3$ in $\mathcal{L}_3$. Lastly, the one-to-one correspondence between loops in $\mathcal{G}_3$ and braids of the three eigenvalues reflects the fact that the fundamental group of $\mathcal{G}_3$ is $B_3$, the Artin braid group[38] on three strands.

This structure is illustrated in Fig. 1a, which shows a stereographic projection of the unit hypersphere $\mathcal{S}_1$, along with the locations of the EPs it contains. The latter can be seen to form a trefoil knot $\mathcal{K}$ (yellow curve). Also shown in Fig. 1a is an example of a 2D subspace $\mathcal{B}$ (gray plane). This particular choice for $\mathcal{B}$ intersects $\mathcal{K}$ at five locations. One of the intersections is tangential (the one at the greatest $Y$ value), while the other four are transverse[39]. Figure 1b depicts $\mathcal{B}$ and the five EPs within it.

To illustrate the specific problem that we consider in the experiments described below, Fig. 1a, b show three control loops. All three loops lie in $\mathcal{B}$ and share a common basepoint. Viewed within $\mathcal{B}$ (as in Fig. 1b), the loops enclose zero, one, and two EPs, and so are not homotopic in $\bar{\mathcal{B}}$. However, it is straightforward to see from Fig. 1a that they are homotopic in $\mathcal{G}_3$. As a result, these three loops all result in the same eigenvalue braid. In fact each of these loops is contractible, and so results in the identity braid (and hence the identity permutation). This agrees with the explicit calculation of $\boldsymbol{\lambda}$ around each loop, shown in Fig. 1c. A detailed description of Fig. 1 is in "Methods".

### Experimental setup
To demonstrate these features experimentally, we used a COM consisting of three vibrational modes of a 1 mm $\times$ 1 mm $\times$ 50 nm Si$_3$N$_4$ membrane. The dynamical matrix $H$ for these modes is controlled by placing the membrane inside a high-finesse optical cavity. When the cavity is driven by one or more lasers, light inside the cavity exerts radiation pressure on the membrane, altering $H$ via the well-known dynamical back-action (DBA) effect of cavity optomechanics[40–42].

Figure 2a shows a schematic of the experiment. The optical cavity is addressed by two lasers, labeled control and probe. The control laser is split into three tones (by an acousto-optic modulator (AOM)), which are used to tune $H$. For each choice of $H$, the membrane's spectrum $\boldsymbol{\lambda}$ is determined from its frequency-dependent mechanical susceptibility

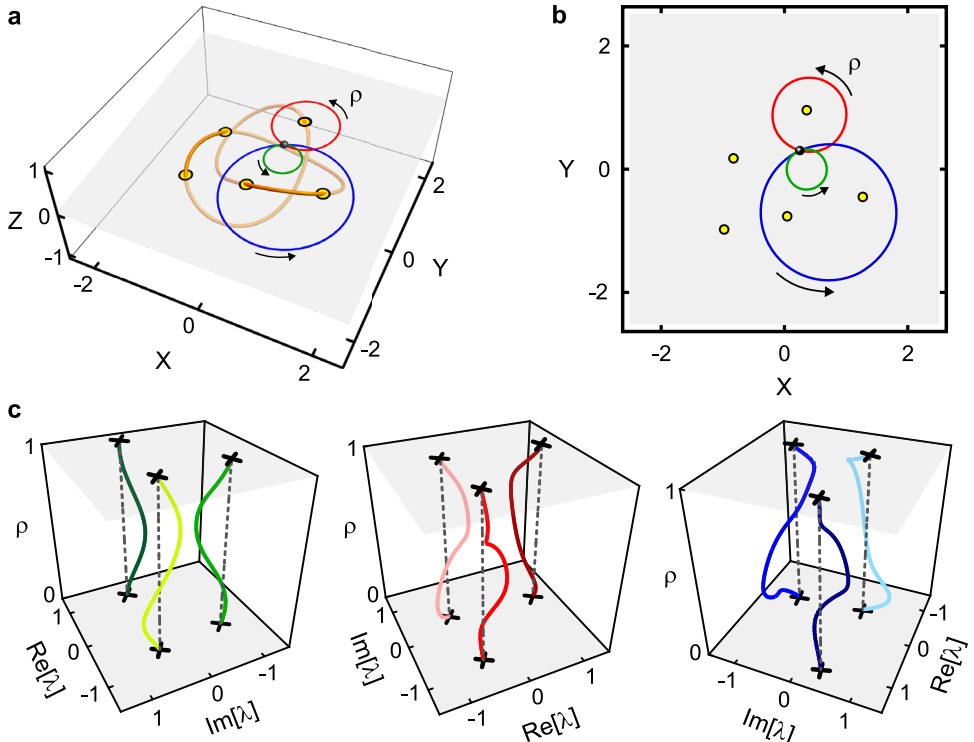

**Fig. 1 | Controlling the spectrum of a three-mode system. a** The full space of control parameters. The EPs are shown in yellow, and form a trefoil knot $\mathcal{K}$. The coordinates $X, Y, Z$ are defined in "Methods". The gray plane shows an example of a 2D subspace ($\mathcal{B}$). The yellow discs are the five intersections of $\mathcal{B}$ with $\mathcal{K}$. The red, green, and blue loops all lie in $\mathcal{B}$ and have a common basepoint (black circle). **b** The control plane $\mathcal{B}$, showing the five EPs it contains (yellow circles) and the three control loops. The coordinate along each loop is $\rho$. For each loop, the basepoint (black circle) corresponds to $\rho = 0$ and $\rho = 1$. **c** The eigenvalue spectrum $\boldsymbol{\lambda}$ calculated as a function of $\rho$ along each of the loops. A detailed description of these plots is "Methods".

$\chi(\tilde{\omega})$, which is measured using the probe laser. A complete description of the setup is given in refs. 18,43.

Figure 2b shows the detuning of the three control tones relative to the cavity resonance. The control tone indexed by $k \in \{1, 2, 3\}$ is detuned by $\sim -\tilde{\omega}_k^{(0)}$ where the elements of $\{\tilde{\omega}_1^{(0)}, \tilde{\omega}_2^{(0)}, \tilde{\omega}_3^{(0)}\}$ are the bare resonance frequencies of the membrane modes in the absence of DBA. For this experiment, we vary the control tones' common detuning $\delta$ (defined in Fig. 2b) and their powers $P_1$, $P_2$, and $P_3$. The relative detunings between the control tones are fixed, and define a rotating frame $\mathcal{R}$ in which the three modes are nearly degenerate for $P_1 = P_2 = P_3 = 0$. As shown in ref. 18, the experimental parameters $\Psi = (\delta, P_1, P_2, P_3)$ provide sufficient control to span $\mathcal{L}_3$ in the neighborhood of the EP$_3$, and in particular to measure the structure of $\mathcal{V}_3$ and $\mathcal{G}_3$.

Figure 2c shows a representative measurement of $\chi(\tilde{\omega})$. The drive is produced by modulating the probe beam's intensity at a frequency $\tilde{\omega}_{AM}$. The resulting heterodyne signal $\bar{V}$, which is proportional to the membrane's motion, is recorded for values of $\tilde{\omega}_{AM}$ in a window centered on $\tilde{\omega}_k^{(0)}$ for each $k$. Figure 2c shows $|\bar{V}|$ in the left panel and a parametric plot of $\bar{V}$ in the right panel for each $k$. This data is fit to the expected form of $\chi(\tilde{\omega})$ using $\boldsymbol{\lambda}$ as a fit parameter[18]. Throughout this paper, $\boldsymbol{\lambda}$ is determined by data and fits such as those shown in Figure 2c, and its values are given in the frame $\mathcal{R}$. Quantities with a tilde are given in the lab frame, and without a tilde in the rotating frame $\mathcal{R}$. The device parameters are given in Table 1.

**Measurements of braids and permutations**

The setup described above was used to measure $\boldsymbol{\lambda}$ at $\sim 3 \times 10^4$ values of $\Psi$[18]. A small subset of this data was taken for values of $\Psi$ ranging throughout $\mathcal{L}_3$ and was used to determine the location of the EP$_3$. The rest of the data was taken for values of $\Psi$ on a hypersurface $\mathcal{S}$ surrounding EP$_3$. These measurements are described in detail in ref. 18.

Figure 3a shows part of this data. At each value of $\Psi$ the measured $\boldsymbol{\lambda}$ was converted to $D = (\lambda_1 - \lambda_2)^2(\lambda_2 - \lambda_3)^2(\lambda_3 - \lambda_1)^2$, which is the discriminant of $p_H$. The color scale in Figure 3a shows $\arg(D)$. The yellow circles show the two locations identified in this data by a vortex-finding algorithm[18]. As described in ref. 18, a vortex in $\arg(D)$ is one signature of an EP$_2$. These locations agree well with those returned by three other algorithmic means of identifying EP$_2$s from the data[18]. Figure 3b displays a stereographic projection of $\mathcal{S}$ in which the EPs identified via vortices in $\arg(D)$ are depicted by the yellow spline curve (see "Methods").

To examine the effect of control loops (viewed in either the full control space or in a 2D subspace) we take the region $\mathcal{B}^{(1)}$ shown in Fig. 3a as our first example of a 2D control space, and consider the two loops shown as green and blue in Fig. 3a. These loops share a common basepoint, and are clearly not homotopic in $\bar{\mathcal{B}}^{(1)}$ (defined as $\mathcal{B}^{(1)}$ without the two EPs). However, Fig. 3b, c show that these loops are homotopic in $\mathcal{G}_3$ and that each loop is contractible. Measurements of $\boldsymbol{\lambda}$ at several positions along each loop are shown in Fig. 3d, e, and demonstrate that these loops produce isotopic braids (in this case, the identity braid), as would be expected from Fig. 3b, c. This illustrates one of the striking features from Fig. 1: the braid traced out by $\boldsymbol{\lambda}$ is determined by the loop's homotopy class in $\mathcal{G}_N$, and not by its homotopy class in any particular $\bar{\mathcal{B}}$.

Another important feature of 2D control spaces is that control loops may encircle the same EPs and yet give rise to distinct eigenvalue braids[16]. This scenario is demonstrated in Fig. 4, which shows the 2D control space $\mathcal{B}^{(2)}$. Here, the two loops (red and blue) share a common basepoint and encircle the same EPs. Nevertheless, they are not homotopic in either $\bar{\mathcal{B}}^{(2)}$ or $\mathcal{G}_3$ (Fig. 4b, c) and the braids they produce (Fig. 4d, e) are not isotopic.

Next, we investigate the role of the loops' basepoints in determining their homotopy equivalence. To do so we use the data shown in Fig. 5, which depicts a third 2D control space $\mathcal{B}^{(3)}$, along with two loops.

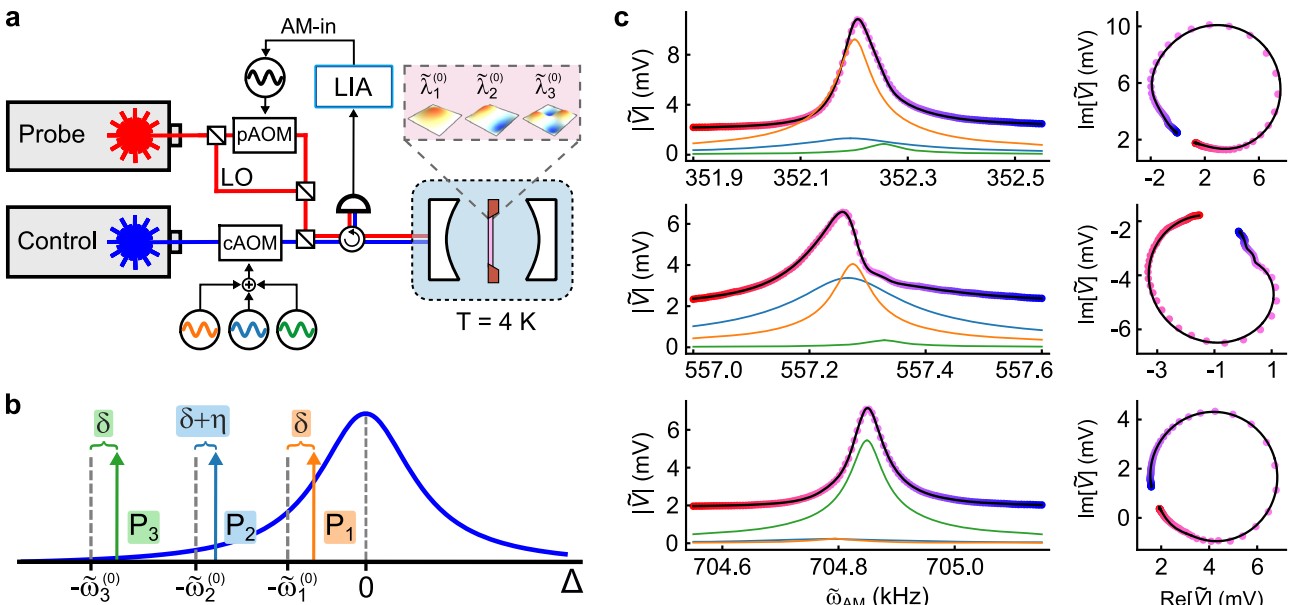

**Fig. 2 | Experimental setup. a** A Si$_3$N$_4$ membrane (red) is placed between the mirrors of a Fabry-Pérot cavity (white) in a cryostat (blue). Three of the membrane's modes (pink box) are tuned using three tones generated from the "control" laser via an AOM (cAOM). The membrane is driven by modulating the "probe" laser's intensity (via the amplitude modulation port (AM-in) of the source that drives a second AOM (pAOM)). The probe laser also provides a local oscillator (LO) which generates a signal $\tilde{V}$ that is proportional to the membrane's displacement and is monitored via a lock-in amplifier (LIA). **b** The detunings $\Delta$ of the three control tones (with respect to the cavity's resonance). Dark blue: the magnitude of the cavity's optical susceptibility. The parameter $\eta = -2\pi \times 100$ Hz is chosen to provide an optimal rotating frame[18]. **c** A measurement of the membrane's mechanical susceptibility for $\Psi = (2\pi \times 46$ kHz, 109.4 µW, 376.8 µW, 77.0 µW). For each frequency range, the left panel shows $|\tilde{V}(\tilde{\omega}_{AM})|$ and the right panel shows a parametric plot of $\tilde{V}$. Each data point is colored according to the value of $\tilde{\omega}_{AM}$. The black lines are a global fit to all the data shown. This fit returns the system's eigenvalues $\boldsymbol{\lambda} = 2\pi \times \{49.670 - i\,84.977, 57.636 - i\,29.834, 112.222 - i\,26.325\}$Hz in the rotating frame $\mathcal{R}$. The magnitude of each mode's contribution (as determined from the fit) is shown as the orange, green, and light blue curves in the left-hand column.

These loops are homotopic if their basepoint is the white circle; however, they are non-homotopic if their basepoint is the black circle. This also holds if the loops are viewed in $\mathcal{L}_3$, as shown in Fig. 5b, c. As a result, the loops generate isotopic braids when based at the white point (Fig. 5d, e) and non-isotopic braids when based at the black point (Fig. 5f, g).

Figure 5 illustrates another feature that is absent for $N = 2$ but generic for $N > 2$: the noncommutativity of braids. Specifically, the braids in Fig. 5f, g do not commute with each other: concatenating the braid in Fig. 5f and the braid in Fig. 5g results in the leftmost eigenvalue

(as it appears in the figure) being transported to the middle one, while reversing the concatenation causes the leftmost eigenvalue to be transported to the rightmost one.

## Discussion

In conclusion, we have considered two complementary ways of describing the topological features that arise when the spectrum of a non-Hermitian system is tuned. The first describes tuning with two control parameters[16], while the second considers all of the system's control parameters[18]. The former offers ease of visualization and corresponds to many actual experiments but may mask the features that determine which control loops are topologically distinct from each other. The latter provides a clear picture of topological equivalence but is more challenging to visualize.

We have illustrated both of these approaches experimentally, using a three-mode mechanical system that is tuned optomechanically. However, we emphasize that the results presented here are generic and can be applied to non-Hermitian systems realized in any physical platform.

These results help to elucidate the complex topology of non-Hermitian systems possessing many degrees of freedom. This insight may aid in efforts to realize new types of mode switching via dynamical encircling of EPs[44–47]. It may also help to clarify which aspects of a linear system's topology remain relevant to its behavior in nonlinear regimes[48,49]. Lastly, it may find application in the fields of non-Hermitian band structure[50–56] and complex scattering phenomena[55,57,58], where closely related concepts arise.

## Methods

### Traacelessness of $H$

Throughout this paper, we take $H$ to be traceless. This choice simplifies the analysis, and does not alter the generality of the conclusions presented here.

## Table 1 | Parameters of mechanical and optical modes, optomechanical coupling, and laser source used for the experiment

| Parameter | Value |
| --- | --- |
| $\tilde{\omega}_1^{(0)}/2\pi$ | 352243.3 ± 0.1 Hz |
| $\tilde{\omega}_2^{(0)}/2\pi$ | 557216.8 ± 0.1 Hz |
| $\tilde{\omega}_3^{(0)}/2\pi$ | 704836.7 ± 0.1 Hz |
| $\tilde{\gamma}_1^{(0)}/2\pi$ | 4.4 ± 0.1 Hz |
| $\tilde{\gamma}_2^{(0)}/2\pi$ | 3.8 ± 0.1 Hz |
| $\tilde{\gamma}_3^{(0)}/2\pi$ | 3.6 ± 0.1 Hz |
| $\kappa/2\pi$ | 173.8 kHz |
| $\kappa_{in}/2\pi$ | 46.4 kHz |
| $g_1/2\pi$ | 0.1979 Hz |
| $g_2/2\pi$ | 0.3442 Hz |
| $g_3/2\pi$ | 0.3092 Hz |
| $\lambda$ | 1064 nm |

$\tilde{\omega}_i^{(0)}$: the bare resonance frequency of the $i$th mechanical mode (i.e., in the absence of any optomechanical effects). $\tilde{\gamma}_i^{(0)}$: the bare energy damping rate of the $i$th mechanical mode. $\kappa$: the optical cavity linewidth. $\kappa_{in}$: the optical cavity input coupling rate. $g_i$: the optomechanical coupling rate between the optical cavity and the $i$th mechanical mode. $\lambda$: the laser wavelength.

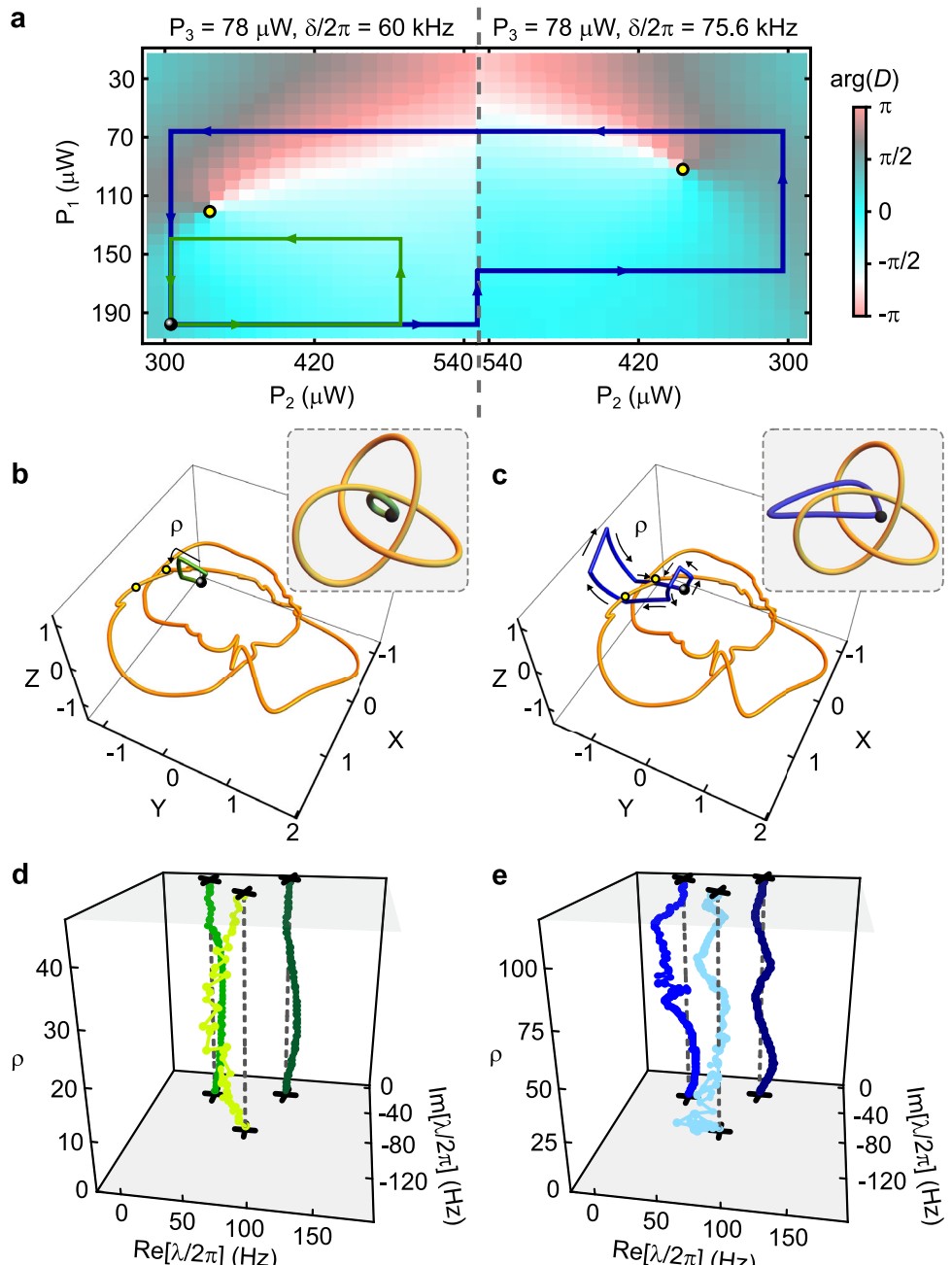

**Fig. 3 | Equivalent eigenvalue braids from different loops. a** The 2D control space $\mathcal{B}^{(1)}$. Color scale: arg($D$). Yellow circles: vortices in arg($D$), which correspond to EPs. Green and blue curves: control loops. Black circle: the loops' basepoint. A view of $\mathcal{B}^{(1)}$ in $\mathcal{L}_3$ is in Supplementary Fig. S1. **b** Stereographic projection of the hypersurface $\mathcal{S}$. The axes $X, Y, Z$ are defined in Methods. Yellow curve: the measured EPs. The black circle, green curve, and yellow circles are as in (**a**). The control loops consist of straight segments in (**a**) but appear curved in (**b**) owing to the stereographic projection. Inset: a simplified cartoon of the relationship between the EPs and the control loop. **c** The same as in (**b**), but for the blue control loop. **d** The eigenvalue spectrum $\lambda$ as a function of position along the green control loop. $\rho$ indexes the measurements of $\lambda$ along the loop. **e** As in (**d**), but for the blue control loop. Although the loops are not homotopic in $\bar{\mathcal{B}}^{(1)}$, they are homotopic in $\mathcal{G}_3$, and so produce isotopic braids.

Intuitively, this can be understood by considering a control loop parameterized by $0 \leq \rho \leq 1$ for which $H(\rho)$ is not assumed to be traceless (and in which tr($H(\rho)$) may vary with $\rho$). The eigenvalues of $H$, viewed as a function of $\rho$, will trace out a braid of $N$ strands[18,20,21]. For each value of $\rho$ along the braid, tr($H(\rho)$) sets the average of the $N$ eigenvalues. If for each value of $\rho$ we remove the trace of $H$ (i.e., replace $H(\rho)$ with $H(\rho) - I$tr($H(\rho)$) where $I$ is the identity matrix), this amounts to translating (in the complex plane) the eigenvalue spectrum as a whole. Even if this translation is different for each value of $\rho$, the fact that it is applied to the spectrum as a whole means that the topological character of the braid is unchanged.

More formally, shifting the center of the eigenvalue braid by an amount that varies with $\rho$ gives a family of homeomorphisms that defines an ambient isotopy (the ambient space of the braids is $\mathbb{C} \times \mathbb{I}$, where $\mathbb{I}$ is the unit interval). As a result, it leaves the braid's isotopy equivalence class unchanged[59].

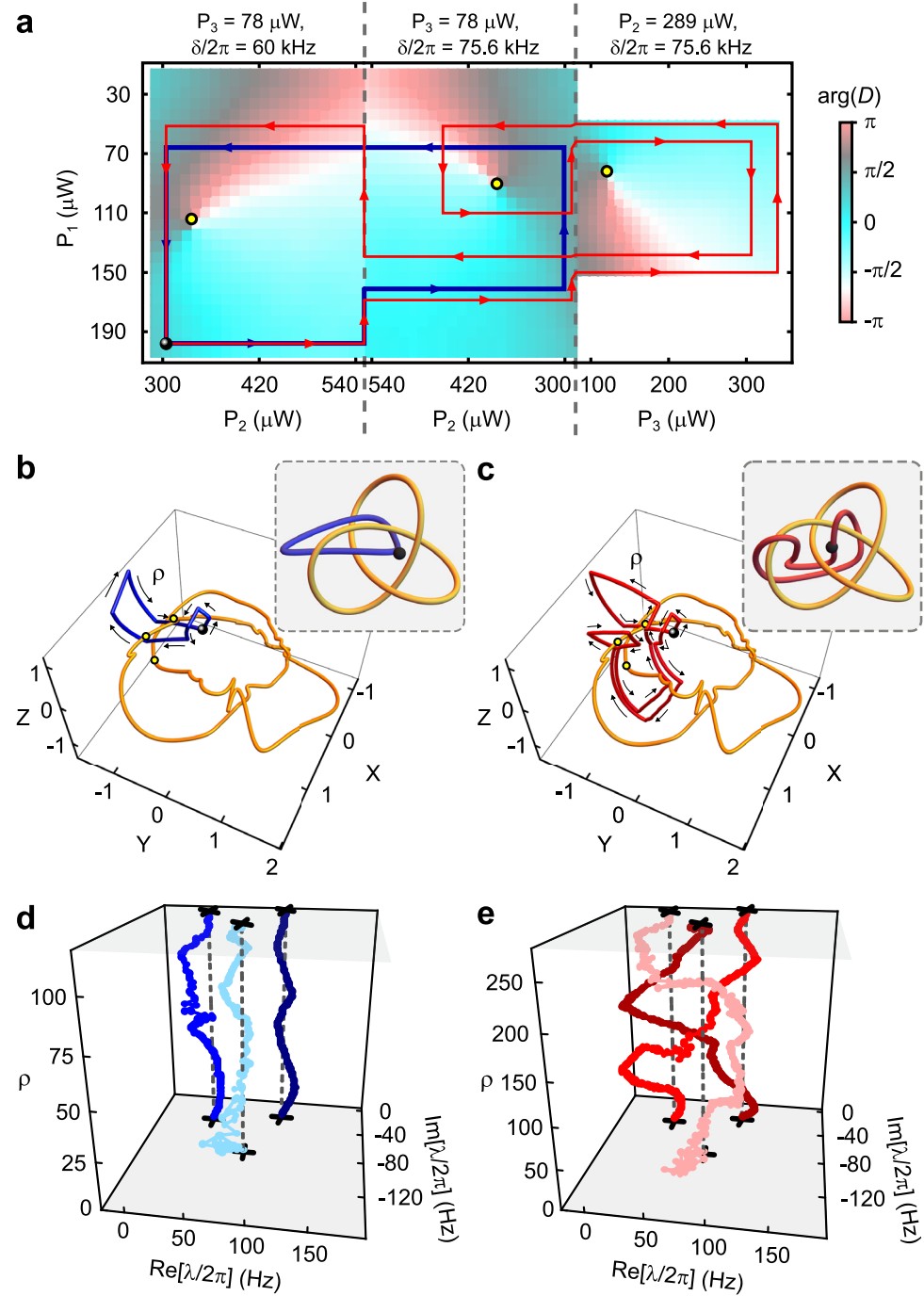

**Fig. 4 | Different braids by encircling the same EPs. a** The 2D control space $\mathcal{B}^{(2)}$. Color scale: arg($D$). Yellow circles: vortices in arg($D$) corresponding to EPs. Blue and red curves: control loops. Black circle: the loops' basepoint. A view of $\mathcal{B}^{(2)}$ in $\mathcal{L}_3$ is in Supplementary Fig. S1. **b** Stereographic projection of $\mathcal{S}$. Yellow curve: the measured EPs. The black circle, blue curve, and yellow circles are as in (**a**). Inset: a simplified cartoon of the relationship between the EPs and the control loop. **c** The same as in (**b**), but for the red control loop. **d** The eigenvalue spectrum $\lambda$ as a function of position along the blue control loop. $\rho$ indexes the measurements of $\lambda$ along the loop. **e** As in (**d**), but for the red control loop. The braids produced by the two loops are not isotopic (and do not produce the same permutation) even though they encircle the same EPs in $\mathcal{B}^{(2)}$.

## Detailed description of Fig. 1

In this section, we provide a detailed description of Fig. 1, which illustrates the space spanned by the coefficients of $p_H$ (the characteristic polynomial of the $3 \times 3$ dynamical matrix $H$). It also shows the structure of the degeneracies within this space, along with one particular choice of a 2D subspace.

The characteristic polynomial of a matrix is monic, meaning that the coefficient of its highest-order term is unity. The coefficient of the next-highest-order term is simply the trace of the matrix. In COMs, the

trace of the dynamical matrix can be trivially absorbed into the definitions of the oscillators' coordinates, so we take tr($H$) = 0 throughout this paper without loss of generality. As a result, the most general form for $p_H$ is $\lambda^3 - y\lambda - x$, where the complex numbers $x$ and $y$ fully specify the eigenvalue spectrum. Degeneracy between roots of $p_H$ corresponds to the condition $D_{p_H} = 0$, where $D_{p_H} = 4y^3 - 27x^2$ is known as the discriminant of $p_H$.

The space spanned by $x$ and $y$ (denoted $\mathcal{L}_3$) can be viewed as $\mathbb{R}^4$ and can be labeled with the Cartesian coordinates

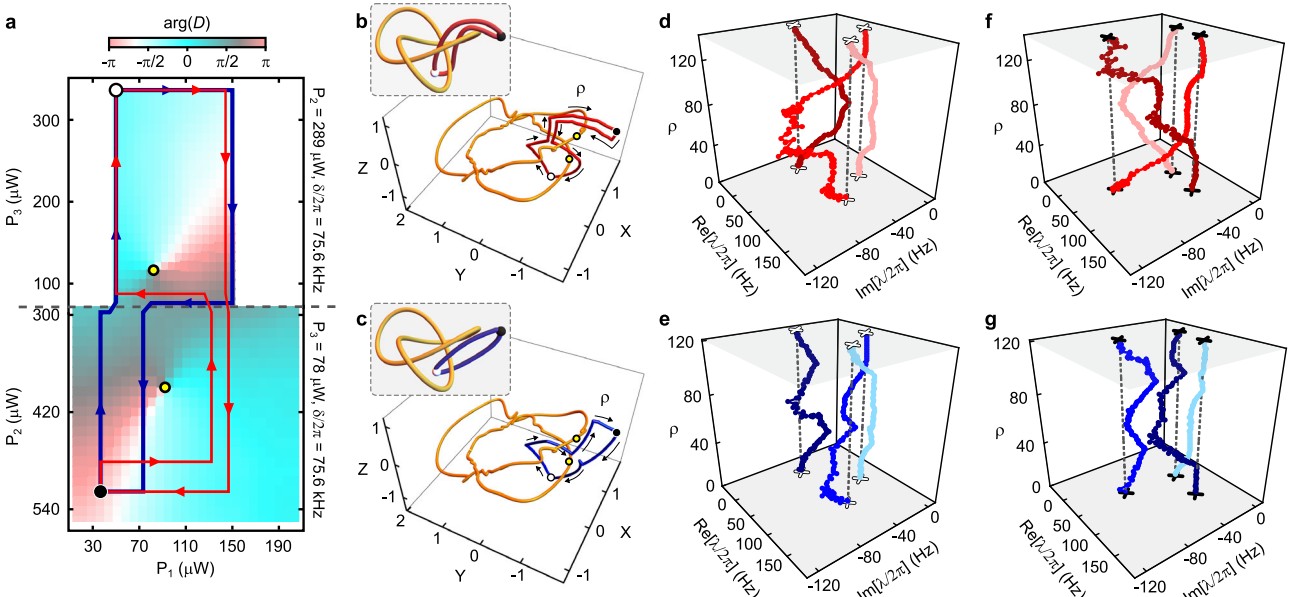

**Fig. 5 | Measuring the basepoint dependence of homotopy equivalence. a** The 2D control space $\mathcal{B}^{(3)}$. Color scale: arg($D$). Yellow circles: vortices in arg($D$) corresponding to EPs. Blue and red curves: control loops. Black and white circles: two choices for the loops' basepoint. A view of $\mathcal{B}^{(3)}$ in $\mathcal{L}_3$ is in Supplementary Fig. S1. **b** Stereographic projection of $\mathcal{S}$. Yellow curve: the measured EPs. The black circle, white circle, red curve, and yellow circles are as in (**a**). Inset: a simplified cartoon of the relationship between the EPs, the control loop, and the basepoints. **c** As in (**b**),

but for the blue control loop. **d** The eigenvalue spectrum $\lambda$ as a function of position along the red control loop, using the white basepoint. $\rho$ indexes the measurements of $\lambda$ along the loop. **e** As in (**d**), but for the blue control loop. **f** As in (**d**), but for the black basepoint. **g** As in (**e**), but for the black basepoint. When the two loops shown in (**a**) are based at the white point, they are homotopic and so produce isotopic braids; however, the same loops based at the black point are not homotopic and produce non-isotopic braids.

(Re($x$),Im($x$),Re($y$),Im($y$)). As described in the main paper (as well as in refs. 18,37,60), any hypersphere centered at the origin of $\mathcal{L}_3$ (i.e., at the point (0, 0, 0, 0)) contains twofold degeneracies (EP$_2$s). These form a closed curve that is a trefoil knot.

Figure 1a shows this structure explicitly. It is a stereographic projection of the unit hypersphere $\mathcal{S}_1$ (defined by $|x|^2 + |y|^2 = 1$) onto $\mathbb{R}^3$. In particular, this projection uses the point $(-1, 0, 0, 0)$ as its pole, so that the coordinates $X, Y, Z$ of Fig. 1 are defined via:

$$\text{Re}(x) = \frac{1 - X^2 - Y^2 - Z^2}{1 + X^2 + Y^2 + Z^2}$$
$$\text{Im}(x) = \frac{2Z}{1 + X^2 + Y^2 + Z^2}$$
$$\text{Re}(y) = \frac{2X}{1 + X^2 + Y^2 + Z^2} \quad (2)$$
$$\text{Im}(y) = \frac{2Y}{1 + X^2 + Y^2 + Z^2}$$

The yellow curve in Fig. 1 shows the locations of the degeneracies within $\mathcal{S}_1$. It is determined by the two constraints $|x|^2 + |y|^2 = 1$ (which defines $\mathcal{S}_1$) and $4y^3 - 27x^2 = 0$ (which defines the vanishing of $D_{p_H}$). It can be seen to form the trefoil knot $\mathcal{K}$.

As described in the main text, there are many settings in which it is useful to consider a 2D subspace of $\mathcal{L}_3$. The choice of such a subspace (denoted $\mathcal{B}$) may be motivated by which pair of parameters are most readily accessible in a particular device. There are infinitely many possible choices for $\mathcal{B}$, so it is helpful to distinguish between features that are generic (i.e., which would result for nearly any choice of $\mathcal{B}$) and features that only result for finely tuned choices of $\mathcal{B}$. For example, a generic $\mathcal{B}$ will have a finite (possibly zero) number of transverse intersections with $\mathcal{K}$, while only fine-tuned choices for $\mathcal{B}$ will have non-transverse intersections with $\mathcal{K}$[39].

Figure 1a shows a particular $\mathcal{B}$ that illustrates both generic and nongeneric features. This $\mathcal{B}$ is defined by $Y + m(Z - Z_0) = 0$, with $m = \tan(-17\pi/36)$ and $Z_0 \approx 0.284$. This $\mathcal{B}$ happens to be a plane in the

coordinates $X, Y, Z$. Its flatness (in these coordinates) is a nongeneric feature, but is chosen to simplify the visualization and does not impact the discussion. This particular $\mathcal{B}$ has four transverse intersections with $\mathcal{K}$. As mentioned above, such intersections are generic, though the specific number depends on the choice of $\mathcal{B}$[39]. It also has one non-transverse intersection, which can be viewed as the result of fine-tuning the parameters $m$ and $Z_0$. This nongeneric intersection is included for illustrative purposes, but is not central to this discussion.

Also shown in Fig. 1a are three control loops, all of which lie in $\mathcal{B}$ and share a common basepoint. Each loop is described by:

$$X(\rho) = X_b + r[\cos(\phi_c) - \cos(2\pi\rho + \phi_c)]$$
$$Y(\rho) = Y_b + r[\sin(\phi_c) - \sin(2\pi\rho + \phi_c)] \quad (3)$$
$$Z(\rho) = Z_0 - Y(\rho)/m$$

where $\rho \in [0, 1]$ parameterizes position along a circular path of radius $r$ that starts and stops at the basepoint $(X_b, Y_b, Z_0 - Y_b/m)$. The angle $\phi_c$ sets the orientation of the circle's center with respect to the basepoint.

The three loops in Fig. 1a all have their basepoint at $X_b = 0.25$, $Y_b = 0.3$, radii $r = \{0.325, 0.6, 1.1\}$, and orientations $\phi_c = \{-7\pi/18, 15\pi/36, -13\pi/36\}$ (for the green, red and blue loop, respectively).

Figure 1b shows the plane $\mathcal{B}$ (i.e., the space spanned by $X$ and $Y$ with $Z = Z_0 - Y/m$), along with the five intersections between $\mathcal{B}$ and $\mathcal{K}$ and the three loops just described. These loops can be seen to enclose zero, one, or two of the five intersections between $\mathcal{B}$ and $\mathcal{K}$.

To plot the eigenvalue braids that result from each of these control loops, we first use Eq. (2) to convert the loop coordinates from $(X, Y, Z)$ as given in Eq. (3) to the coordinates (Re($x$),Im($x$),Re($y$),Im($y$)). We then find the roots of $p_H$ numerically for 101 values of $\rho$ ranging from 0 to 1. For each value of $\rho$, we show the three roots (which comprise the eigenspectrum $\lambda$ of $H$) in the complex plane (indicated by Re($\lambda$) and Im($\lambda$) in Fig. 1c). Their evolution as a function of $\rho$ is shown in Fig. 1c by stacking a copy of the complex plane for each $\rho$. The black

crosses highlight $\boldsymbol{\lambda}$ at $\rho = 0$ (the bottom of each plot), which by construction is identical to $\boldsymbol{\lambda}$ at $\rho = 1$ (the top of each plot).

## Representing the measured knot of degeneracies

The features shown in Fig. 1 are calculated using a general 3-mode COM, while Figs. 3b, c, 4b, c, and 5b, c show the corresponding features measured in the optomechanical system described in the main text (and in greater detail in ref. 18). In the experiments described here, $H$ is tuned via the parameters $(\delta, P_1, P_2, P_3)$ defined in the main text. As described in ref. 18, these parameters span $\mathcal{L}_3$ in the neighborhood of $EP_3$. Specifically, the Jacobian relating $(\delta, P_1, P_2, P_3)$ to $(\text{Re}(x), \text{Im}(x), \text{Re}(y), \text{Im}(y))$ has non-zero determinant.

Degeneracies of $H$ were located by fixing two of these parameters (for example, $\delta$ and $P_1$) and densely rastering the other two (in this example, $P_2$ and $P_3$). At each point in this "sheet", the mechanical susceptibility $\chi(\tilde{\omega})$ was measured for three windows of $\tilde{\omega}$ centered near $\tilde{\omega}_1^{(0)}$, $\tilde{\omega}_2^{(0)}$, and $\tilde{\omega}_3^{(0)}$. A typical example of such a measurement is shown in Fig. 2c. These data are fit to the expected form of the mechanical susceptibility, which is the sum of nine Lorentzians.

The fit returns 13 complex fit parameters. First, there are the complex frequencies of the three modes; these comprise $\boldsymbol{\lambda}$. (The center frequencies of the other six Lorentzians are fixed by these values and the frequencies of the intracavity beatnotes produced by the control laser tones). Next, there is a complex constant background and a transduction factor in each of the three windows. Lastly, there are the nine complex amplitudes $s_{i,j}$ of the Lorentzians (denoting the amplitude of the $i$th Lorentzian in the window near $\tilde{\omega}_j^{(0)}$); however, only four of the $s_{i,j}$ are linearly independent. A detailed description of the fitting procedure is in ref. 18.

The quantity $D = (\lambda_1 - \lambda_2)^2 (\lambda_2 - \lambda_3)^2 (\lambda_3 - \lambda_1)^2$ is calculated directly from the value of $\boldsymbol{\lambda}$ returned by this fit. Figures 3a, 4a, and 5a, all show measurements of $\text{arg}(D)$.

To locate the degeneracies of $H$, we make use of $D$ as well as the quantity $E \equiv (\det(S))^{-2}$ where $S$ is the matrix whose elements are the $s_{i,j}$. Both $D$ and $E$ are complex numbers. Both are expected to vanish at an EP, and to exhibit a $2\pi$ phase winding around an EP. As a result, the EPs within a given sheet were identified in four ways: by applying a minimum-finding algorithm to both $|D|$ and $|E|$, and by applying a vortex-finding algorithm to both $\text{arg}(D)$ and $\text{arg}(E)$[18]. This procedure was repeated for sixty-one distinct sheets, all lying in a hypersurface $\mathcal{S}$, and resulted in the identification of 291 degeneracies.

As described in ref. 18, $\mathcal{S}$ consists of eight 3D rectangular cuboids ("faces"). Each 3D face corresponds to fixing one of the four experimental control parameters $(\delta, P_1, P_2, P_3)$ to its minimum (or its maximum) value, and allowing the other three parameters to range from their minimum to maximum values.

Figures 3b, c, 4b, c, and 5b, c, all show the same stereographic projection from $\mathcal{S}$ to $\mathbb{R}^3$. The details of this projection are given in ref. 18. These figures also show the experimentally determined EPs within $\mathcal{S}$. In ref. 18, all of the 291 EPs were displayed as individual points (e.g., in Fig. 3 of that paper). For ease of visualization, in the present paper we replace the points with a curve. This curve is a cubic spline through the 67 degeneracies that were identified as vortices in $\text{arg}(D)$. Producing such a curve requires not only the locations of the points, but also their ordering (i.e., it is necessary to know which degeneracies are "next to" each other along the knot). Since $\mathcal{K}$ is a smooth curve, this ordering could in principle be inferred directly from the locations of the degeneracies (i.e., from their coordinates $X, Y, Z$). However, with a finite number of experimentally determined degeneracies this approach can lead to ambiguities, for example when distinct portions of the curve $\mathcal{K}$ happen to pass close to each other. To remove this potential ambiguity, we note that each degeneracy is also characterized by a parameter $\theta$ which runs from 0 to $2\pi$ around $\mathcal{K}$. The formal definition of $\theta$ is given in ref. 18, but here we note that $\theta$ is simply the complex argument of the non-degenerate member of $\boldsymbol{\lambda}$ (when viewed

in a rotating frame in which $H$ is traceless), and so is readily determined from the data.

To place the experimentally identified degeneracies in order (i.e., as they appear around $\mathcal{K}$), we define for the $i$th point the quantity $\Omega_i = (X_i, Y_i, Z_i, \cos[\theta_i], \sin[\theta_i])$. Then for each point, we find the nearest (and the next nearest) neighbor based on the Euclidean distance between the various $\Omega_i$. Using this information, we sort the list of degeneracies so that the nearest neighbors are the adjacent elements of the list. We then interpolate the sorted list with a cubic spline.

## Description of the three 2D control spaces

This section describes the three 2D control spaces used in the main text. Each of these consists of the union of two or three of the "sheets" described in the previous section (and in ref. 18).

The 2D control space shown in Fig. 3 ($\mathcal{B}^{(1)}$) is the union of two sheets. In the first, $P_3$ and $\delta$ are fixed to 78 $\mu$W and $2\pi \times 60$ kHz respectively. In the second, $P_3$ and $\delta$ are fixed to 78 $\mu$W and $2\pi \times 75.6$ kHz, respectively. The 2D control space shown in Fig. 4 ($\mathcal{B}^{(2)}$) is the union of three sheets. The first two are the same as for $\mathcal{B}^{(1)}$, and the third consists of fixing $P_2$ and $\delta$ to 289 $\mu$W and $2\pi \times 75.6$ kHz, respectively. The 2D control space shown in Fig. 5 ($\mathcal{B}^{(3)}$) is the union of two sheets: the latter sheet from $\mathcal{B}^{(1)}$ and the lattermost sheet from $\mathcal{B}^{(2)}$.

The three sheets used to form $\mathcal{B}^{(1)}$, $\mathcal{B}^{(2)}$, and $\mathcal{B}^{(3)}$ are shown in Supplementary Fig. 1. One of the views in Supplementary Fig. 1 uses the same stereographic projection as in Figs. 3b, c, 4b, c, and 5b, c. The other view uses the "rectilinear stereographic" projection described in ref. 18. The latter provides easier interpretation in terms the experimental parameters $(\delta, P_1, P_2, P_3)$.

## Device parameters

This section gives the values of the various parameters in the optomechanical model of the experimental three-mode system. The model is described in detail in ref. 18. The values (presented in Table 1) are determined by standard optomechanical characterization measurements as described in refs. 18,43.

## Scaling the experiment to larger N

In principle, the experimental approach described here can be extended to any number of modes. The membrane hosts many mechanical modes that couple optomechanically to the cavity, and in general any number $N$ of these can be tuned in the manner used here. Specifically, the $N-1$ coefficients of the characteristic polynomial of these modes' dynamical matrix can be varied over the complex plane via the detuning and power of $N$ laser tones. Roughly speaking, each tone is detuned from the cavity resonance by an amount $-\omega_n + \delta$, where $\omega_n$ is the frequency of the $n$th mechanical mode and $\delta$ is a single parameter common to all of the tones. In this picture, the beatnote between any two of these tones results in coupling between the corresponding mechanical modes.

This specific approach to tuning a non-Hermitian system is generic to any situation in which the $N$ modes all parametrically couple to an auxiliary mode that can be driven externally. Tracing out this auxiliary mode from the equations of motion leaves an $N$-mode non-Hermitian system whose parameters are determined by the drive tones applied to the auxiliary mode. In the work presented here, the $N$ modes are the membrane's mechanical modes and the auxiliary mode is the optical cavity.

In practice, a number of issues may limit the maximum value of $N$ that can be tuned in this way. For example, if the ratio $\omega_n/\kappa$ (where $\kappa$ is the cavity linewidth) is too small or too large, it becomes challenging to tune the matrix elements of $H$ over a large region of the complex plane. In addition, complications arise if multiple mode pairs share the same frequency difference, as this means that there is no simple correspondence between the laser beatnotes and the mechanical modes between which they induce coupling.

## Data availability

The data generated in this study have been deposited in the Zenodo database under accession code https://zenodo.org/records/10451386.

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

## Acknowledgements

This work is supported by Air Force Office of Scientific Research (AFOSR) Multidisciplinary University Research Initiative (MURI) Award on Programmable Systems with non-Hermitian quantum dynamics, award No. FA9550-21-1-0202 (R.E., S.K.O., and J.G.E.H); the Alexander von Humboldt Foundation (R.E.); AFOSR award no. FA9550-18-1-0235 (S.K.O.); AFOSR award no. FA9550-15-1-0270 (J.G.E.H.); and Vannevar Bush Faculty Fellowship no. N00014-20-1-2628 (J.G.E.H.). We thank P. Henry, J. Höller, L. Jiang, N. Kralj, J. R. Lane, N. Read, and Y. Zhang for helpful conversations.

## Author contributions

J.G.E.H. and R.E. conceived the project. Y.S.S.P. supervised the data acquisition and analysis. C.G. and Y.S.S.P. performed the experiment and analyzed the data with feedback from Q.Z. and S.K.O. All authors contributed to the manuscript preparation.

## Competing interests

The authors declare no competing interests.
