## [Peer Review File · Nature Communications]

REVIEWER COMMENTS

Reviewer #2 (Remarks to the Author):

Report for manuscript NCOMMS-23-18541-T

Title: Resolving the topology of encircling multiple exceptional points

In this work the authors investigate the eigenvalue braiding occurring in a non-Hermitian system with a spectrum that exhibits multiple exceptional points. Contrary to recent studies that focus on the band braiding of periodic non-Hermitian systems, the present work considers a finite structure made of three coupled acoustic resonances using a laser-controlled nano-membrane and investigates in detail the relationship between the eigenvalue braiding and the topology of the encircling loops.

What I find particularly interesting is that the authors consider the same problem from two different points of view. The first of which treats the encircling as viewed from a plane that contains the encircling loops. In this case, the non-Hermitian singularities appear as isolated points. The other perspective considers an enlarged parameter space and in this case the singularities form a knot in 3D space. As the authors argue, the first perspective provides an easier way for characterizing the system and extracting some of its interesting features. These include for instance the dependence of the braiding results on the starting point on the loop, and the possible inequivalence between two loops even when they encircle the same exceptional points. These results are pertinent only to systems with multiple exceptional points and do not exist in simpler situations where only a single exceptional point is involved. On the other hand, the perspective from a higher dimension space allows one to understand some results that may appear totally accidental at first sight. The authors demonstrate this by considering a situation where some loops in 2D space may seem to be inequivalent yet give the same braiding results. As it turned out, these loops are topologically equivalent in the enlarged 3D space (i.e., they can be deformed into one another without crossing the knot of exceptional points).

The manuscript is well-written and the theoretical and experimental results are clearly explained. This work provides a valuable insight into the physics of non-Hermitian systems in more complex setups than those studied previously. For these reasons I consider the manuscript suitable for publication in Nature Communications.

I just have a few questions that the authors should consider:

1. Can the authors comment on the scalability of their system? In other words, how easy/difficult will it be to extend the system to even larger number of modes with more exceptional points?

2. The current study considers only topology of the eigenspace. Is it possible to extend this to dynamic encircling of exceptional points in future work?
3. The authors do not explicitly investigate the non-abelian nature of the braiding here, but I assume it is implied since they deal with more than two states. Can the authors briefly comment on that aspect?
4. In the Methods section, the authors use a quantity denoted by E but refer the readers to Ref. [13] for the definition. In my opinion, this is not the best practice. I would strongly recommend redefining this quantity in the current manuscript.
5. When I look at Fig.3(a), I see that the figure is split into two domains separated by a dashed line. In each domain the encircling trajectory is varied across two parameters, i.e., it lies in a plane. But when I look at the loops in Figs. 3 (b) and (c), the loops look curved at each point. This discrepancy needs some clarification.

Reviewer #3 (Remarks to the Author):

This is an interesting important contribution to the field by experimentally showing that richer behavior occurs in systems with more than two modes. In their abstract they emphasized the novelty of their work "To date, most work has focused on systems consisting of two modes that are tuned by two control parameters and have isolated EP". However, they oversimplified the current situation when two parameters are varied to get an asymmetric switch

For example,

see Pick, A. et al in (2019), Robust mode conversion in nv centers using exceptional points. Phys. Rev. Res., 1(1), where "While previous theoretical and experimental work on EP-based mode switches applies only to pure states, we develop here a general theory for switching between mixed states, statistical ensembles of different pure states, resulting from the interaction with the environment. Our theory is general and applicable to all leading platforms for quantum information processing and quantum technologies. However, our numerical simulations use empirical parameters of NV centers. We provide guidelines for coping with the main challenges for experimental realization of this protocol: decoherence and mixed-state preparation".

see Guy Elbaz et al paper on Encircling exceptional points of Bloch waves: mode conversion and anomalous scattering. J. Phys. D: Appl. Phys. 55 235301 (2022), where "We investigate how, collectively, mode conversion in the laminate and the fields it scatters depend on the parameters of the loop. We find that the starting point of the loop has a significant effect on various counterintuitive phenomena: it determines if the laminate acts as a source or a sink of energy; how mode conversion takes place; if the reflectance is greater than one; and if there is spatial asymmetry in the energy flow with respect to the

direction of the incident waves. Our findings are relevant for the development of devices for elastic wave manipulation"

see Choi et al paper on A Principle of Non-Hermitian Wave Modulators by Indefinitely Small Physical Controls. Laser Photonics Rev 2023, 2200580, where "Interferometers and resonant cavities are indispensable driving mechanisms for compact, high-speed, and low-power modulators and switches in modern signal processing systems. However, their limitations in key performance metrics critically restrict present data-processing capabilities. Here, a completely different wave-modulation mechanism is proposed based on non-Hermitian dynamics near an exceptional point (EP) singularity. The proposed modulator is enabled by EP-bypassing adiabatic processes that exclusively select different final states depending on active trigger signal possibly at indefinitely small magnitude in principle. Importantly, this operation principle does not involve any explicit frequency-dispersive feature in stark contrast to interference or resonance effects. In addition, it can be implemented in available device-engineering platforms such as integrated optical circuits."

Last but not least the authors have forgotten to give a reference to the first work to introduce the asymmetric dynamics when parameters are varied in time to encircle EP

Uzdin, R. et al paper in (2011). On the observability and asymmetry of adiabatic state flips generated by exceptional points. J. Phys. A-Math. Theor., 44(43)

I recommend publication after minor revision where the above points are addressed

Response to Reviewer #2:

Regarding the five specific questions raised by the referee:

1. Can the authors comment on the scalability of their system? In other words, how easy/difficult will it be to extend the system to even larger number of modes with more exceptional points?

In principle, the experimental approach described here can be extended to any number of modes. The membrane hosts many mechanical modes that couple optomechanically to the cavity, and in general any number N of these modes can be tuned in the manner described in our paper. Specifically, the $N-1$ coefficients of the characteristic polynomial of these modes' dynamical matrix can be varied over the complex plane by varying the detuning and power of N laser tones. Roughly speaking, this is accomplished by having each tone detuned from the cavity resonance by an amount $-\omega_n + \delta$, where ω_n is the frequency of the n th mechanical mode and δ is a single parameter common to all of the tones. In this picture, the beatnote between two of the tones results in coupling between the corresponding mechanical modes.

This specific approach to tuning a non-Hermitian system is generic to any situation in which the N modes are all parametrically coupled to a single auxiliary mode that can be driven externally. Tracing out this auxiliary mode from the equations of motion leaves an N -mode non-Hermitian system whose parameters are determined by the driving applied to the auxiliary mode. In the case of the present paper, the N modes are the membrane's mechanical modes and the auxiliary mode is the optical cavity mode.

In practice, a number of issues may limit the maximum value of N that can be tuned in this way. For example, if the ratio ω_n / κ (where κ is the cavity linewidth) is too small or too large, it becomes challenging to tune the corresponding matrix elements of H over a large region of the complex plane. In addition, complications arise if multiple pairs of ω_n share the same frequency difference, as this means that the laser beatnotes do not have a simple correspondence with the mechanical modes.

We have added a discussion of these points to the Methods section of the paper.

2. The current study considers only topology of the eigenspace. Is it possible to extend this to dynamic encircling of exceptional points in future work?

As the reviewer states, our work is on the topological structure of the eigenvalue spectrum. This structure has consequences for the system's dynamics (for example when encircling the EPs in real time). However this relationship is rather complicated, owing for example to the absence of an adiabatic limit for most such operations. As a result we have kept the focus of this paper on the topological structure itself, with explicit connections to real-time dynamics given in the references (for example Refs. 44 – 47 in the Conclusion paragraph, and the references in the Introductory section).

3. The authors do not explicitly investigate the non-abelian nature of the braiding here, but I assume it is implied since they deal with more than two states. Can the authors briefly comment on that aspect?

We have added a short paragraph on this point (beginning: “Figure 5 illustrates another feature...”) just before the conclusion. Specifically, we point out that the braids in Fig. 5(f,g) do not commute, and that this is a generic feature of systems with $N > 2$.

4. In the Methods section, the authors use a quantity denoted by E but refer the readers to Ref. [13] for the definition. In my opinion, this is not the best practice. I would strongly recommend redefining this quantity in the current manuscript.

We agree with this point, and have added a description of the quantity E to the Methods section (it is in the section “Representing the measured knot of degeneracies”).

5. When I look at Fig.3(a), I see that the figure is split into two domains separated by a dashed line. In each domain the encircling trajectory is varied across two parameters, i.e., it lies in a plane. But when I look at the loops in Figs. 3 (b) and (c), the loops look curved at each point. This discrepancy needs some clarification.

In Fig. 3(a), the loop is comprised of straight segments. However these segments appear curved in (b) and (c) because of the stereographic projection. A point clarifying this has been added to the caption of Fig. 3.

Response to Reviewer #3:

The reviewer mentions four papers:

(A) Pick et al., which considers a “9 mode” non-Hermitian system (in the language of our manuscript; the 9 modes of Pick et al. correspond to the density matrix of a 3-level open quantum system) that are tuned by 4 control parameters. This paper considers both the parametric dependence of the spectrum (as in our manuscript) and the real-time dynamics of the system during a control loop.

(B) Elbaz et al., which considers the scattering matrix of a resonant structure coupled to an infinite space (rather than the “Hamiltonian” or dynamical matrix of a finite number of modes). The system is tuned by 3 parameters. This paper considers both the parametric dependence of the spectrum (as in our manuscript) and the real-time dynamics of the system during a control loop.

(C) Choi et al., which considers a promising device involving two modes and two control parameters. This paper considers both the parametric dependence of the spectrum (as in our manuscript) and the real-time dynamics of the system during a control loop.

(D) Uzdin et al., which is an important early work describing the non-reciprocal outcomes of dynamic encircling.

We agree with the reviewer that these are important papers and should be cited. Since each paper relates somewhat differently to the topic of our manuscript, we have included them as follows.

In the second paragraph of the paper (beginning: “One such feature...”) which is part of the introduction, we have included references both to recent reviews on non-Hermitian systems (1-6) and to representative early papers on the topology of the eigenvalue spectrum (7 – 13). The paper (D) mentioned by the reviewer is included in the latter.

In the last paragraph of the introduction (beginning: “The rich behavior...”) we have included references to work on systems with more than 2 modes, including the paper (A) mentioned by the reviewer.

We have expanded the final paragraph (beginning: “These results help...”), which discusses the broader implications of the current work. Specifically, we have added a comment on the potential connections to other settings that involve complex matrices (like scattering matrices that describe the coupling of linear systems to external waves) and have included there a reference to the paper (B) mentioned by the reviewer. This paragraph now also includes a reference to paper (C) along with others that discuss “...new types of mode switching via dynamical encircling of EPs.”

Summary of changes to the manuscript:

1) We have added a short paragraph (“Figure 5 illustrates another feature...”) in response to the question raised by Reviewer #2.

2) We have added a short sentence to the caption of Fig. 3 to explain the apparent curvature of the loops, in response to the question from Reviewer #2.

3) In Methods, we have added text to the section “Representing the measured knot of degeneracies” that explains the parameter “ E ”, as requested by Reviewer #2.

4) In Methods, we have added a section “Scaling the experiment to larger N ” to address the question raised by Reviewer #2.

5) We have added references 55, 47, 12, 36 in response to Reviewer #3. We have also added references 56, 57, 58, 7, 13 as they are closely related. In the final paragraph of the paper, we have mentioned that the features of non-Hermitian matrices studied here are also relevant to the study of scattering problems.

REVIEWERS' COMMENTS

Reviewer #2 (Remarks to the Author):

The authors have done a very good job addressing all my points from the first review round. As a result, I recommend publication of the paper.

Reviewer #3 (Remarks to the Author):

The modified paper is now suitable for publication in Nature Communication.

I recommend to accept it for publication.